# Three-Axis Tension-Measuring Vitreoretinal Forceps Using Strain Sensor for Corneal Surgery

**DOI:** 10.3390/polym13244433

**Published:** 2021-12-17

**Authors:** Seongjin Yang, Suhyeon Kim, Seong Kyung Hong, Hyungkook Jeon, Seong J. Cho, Geunbae Lim

**Affiliations:** 1Mechanical Engineering, Pohang University of Science and Technology (POSTECH), 77 Cheongam-ro, Nam-gu, Pohang 790-784, Korea; seongjin@postech.ac.kr (S.Y.); kshyeon@postech.ac.kr (S.K.); skhong@postech.ac.kr (S.K.H.); skyblue@postech.ac.kr (H.J.); 2School of Mechanical Engineering, Chungnam National University, 99 Daehak-ro, Yuseong-gu, Daejeon 305-764, Korea

**Keywords:** flexible sensor, healthcare monitoring, MEMS, stretchable electronics

## Abstract

Precise motion control is important in robotic surgery, especially corneal surgery. This paper develops a new tension-measurement system for forceps used in corneal surgery, wherein contact force is applied only to a specific location for precise control, with precise movements detected by attaching a nano-crack sensor to the corresponding part. The nano-crack sensor used here customizes the working range and sensor sensitivity to match the strain rate of the tip of the forceps. Therefore, the tension in the suture can be sufficiently measured even at suture failure. The printed circuit board attached to the bottom of the system is designed to simultaneously collect data from several sensors, visualizing the direction and magnitude of the tension in order to inform the surgeon of how much tension is being applied. This system was verified by performing pig-corneal suturing.

## 1. Introduction

In the case of corneal transplantation surgery, sophisticated suturing is essential to improve accuracy since failure is always a possibility due to infection or astigmatism [1]. In addition, any minor damage to the cornea may result in the rupture of weak scarring at the suture border. In other words, the quality of corneal surgery is entirely dependent on the skill and experience of the surgeon in question. Clearly, problems can often be the result of human error, such as applying excessive force or the slip of a hand, both of which can negatively affect the success of the surgery. In order to address such problems, research is currently being conducted with respect to automated robot surgery [2,3,4,5,6]. In addition, researches such as fiber-bragg (FBG) based corneal surgical forceps are being made to increase accuracy, but due to shape and size limitation, it does not give sufficient feedback on whether accurate movements were made when applied to an automated robot surgery [7,8,9]. Therefore, automated robot surgery removes human error, it currently lacks the precise movements required for corneal surgery. In other words, automated robot surgery is not without its own problems. These are associated with force feedback, grasping force and slip point [9,10,11], and force-sensing systems [12,13,14,15,16,17,18,19,20]. Indeed, force-sensing systems are currently being studied by many researchers. An example of one of the issues that cause inaccuracy in applications such as corneal surgery is the difficulty of measuring the direct contact force. In most cases, research focuses on bulk gripper tools. In this case, a small contact sensor is inserted in the gripper to measure the grasping contact force. Accordingly, the size of the gripper necessarily increases. In addition, rather than measuring the grasping force, it should be possible to determine whether the suture in question is broken by the tension on the corneal suture when pulling the suture during surgery. This requires a small and precise gripper-measurement system, as well as precise force measurements. Moreover, this system should measure and provide tension information in real time. The purpose of this study is to develop a forceps capable of precise movement and a system that can measure the tension in the range where the suture does not break in real time. Indeed, in order to be applied to corneal surgery, it is necessary to have a particularly miniaturization and unusual structure application, and a sensor that is stretchable and sensitive to low force is required. Unfortunately, as the sensor decreases in size, the required specifications change. Moreover, current tensile-measurement sensors, such as strain gauges, have many problems associated with size, unusual structure application and trade-off between sensitivity and stretchability. Also with respect to commercial sensors, additional measurement equipment is often required because they predominantly use Wheatstone bridges. In general, accurate measurement of the resistance change of the sensor requires Wheatstone bridge-based equipment with three resistors, but this is an obstacle to the need to be miniaturized. Accordingly, research is currently being conducted with respect to flexible strain sensors. Many studies have focused on flexible materials in various fields, including semiconductors and robotic engineering. In the field of strain sensors, ultra-sensitive sensors using nano-cracks have been developed, but these are vulnerable to external damage and also have a small working range. This limits their practical application, since in many cases they focus exclusively on ultra-sensitivity [21]. Recently, interest in human-friendly electronic devices has increased and, accordingly, there is a need for developing sensors with appropriate specifications [22]. Therefore, flexible strain sensors should be developed that can be applied to miniaturized sensors. In this paper, we explain how we have developed devices that can hold and operate micro sutures as well as miniaturized strain sensors using technology inspired by nature. Based on these, we have developed a real-time three-axis tension-measuring forceps system and a sensor that can be used for a number of applications due to its flexibility and high sensitivity. This sensor is advantageous because it can be designed according to desired specifications. By adjusting the surface nano-cracks to the desired size, we can control the sensitivity and the tensile range [23,24,25,26], which, in turn, allows designers to tailor the performance of the sensor according to the specifications required by surgery. The sensor developed through this can be applied to various types of structures and is designed to fit the range of forces used in corneal surgery. Moreover, it has relatively minimal associated costs, a broad working range, and does not require additional equipment [23]. This device has been verified by a corneal ex-vivo test. The ability to grip sutures hundreds of micrometers in diameter is important in corneal surgery. Accordingly, we designed the forceps to accurately hold fine sutures and developed a system that converts tension into resistance data. The resistance data collected from each sensor can be converted into strain and visualized in real time. In this way, it is possible to measure the tension acting on the suture. Accordingly, these forceps are expected to play a vital role both in improving accuracy and reducing failure during surgery. In addition, a calculation method that combines data collected from multiple sensors with high quality can also be utilized [27].

## 2. Design and Methods

### 2.1. Overview of Forceps Sensor System

A schematic of the forceps sensor system is shown in Figure 1. The entire system consists of the following three parts: a suture gripping part, which is deformed when upper tension is applied, a driving body, and a printed circuit board (PCB) for data collection. Of these, a fine nano-crack-based strain sensor is attached to the shaft in order to measure the deformation caused by the suture tension. Corneal surgery requires precise control of surgical instruments and fine control of the force acting on the suture in order to avoid breakage. Therefore, when pulling a thin suture, it is necessary to be able to measure shear-tension changes. Generally, the breaking force of the thinnest suture used in the experiment is roughly 0.3 N. Accordingly, the device should be able to hold thin corneal sutures as well as to measure the deformation caused by suture tension. From a kinematic point of view, the forceps should be able to hold 100-um-diameter sutures without exceeding a tension force of 0.3 N; moreover, a micro-strain sensor is required to detect deformation. In corneal surgery, tension is applied to the suture by the forceps. When tension is applied to the suture, stress is applied to the shaft of the forceps by the tension, and accordingly, the shaft is bent and deformed. The tension applied to the suture can be calculated on the principle that a sensor attached to the shaft surface measures the degree of this deformation. At this point, the resistance-change data is collected from the PCB and the strain relating to each direction can be calculated. This data can be used to generate a graph or as a real-time warning system to prevent the surgeon exceeding a predetermined tension limit.

### 2.2. Design of Nano-Crack-Based Sensor for Smart Vitreoretinal Forceps

It is important to design an entire forceps system to measure the suture tension during corneal surgery as well as to create a sensor with the desired performance and application range. Therefore, a highly sensitive and flexible sensor that can be attached to various curves is required in order to measure shaft deformation at the end of the device when a fine tensile force of 0.3 N or less is applied. In more detail, the suture for corneal surgery should be well-stretched even under a force of less than 0.3 N, which is the corneal surgical suture breaks, and it should be able to measure motion in a microscopic deformation of less than 3% sufficiently. However, the majority of strain gauges and sensors are polyimide-based strain-gauge devices, which have limitations with respect to various non-planar surfaces and applications. In addition, because these types of devices are based on Wheatstone-bridge circuits, additional measurement equipment to more precise measurement is required. Existing commercial strain sensors are large and not flexible enough to be attached to forceps used for corneal surgery, and the range of target force of sensor is much larger than the required range. To satisfy this, a sensor that is more sensitive enough to measure the stretch of the suture and thin and flexible enough to be attached to surgical forceps is required. In order to measure a force of 0.3 N or less, a sensor capable of designing the measuring range and sensitivity according to the purpose is required, and the nano-crack sensor makes this possible. Considering these characteristics, a durable and flexible sensor that is sensitive enough to measure the deformation of the polymer tip is required for corneal surgery.

Indeed, nano-crack-based strain sensors, which simulate the nano-crack structure of spider joints, can be used to address this problem. In order to make a nano-crack sensor, it is necessary to control the crack size of the sensor surface. The size of the crack depends on the method, time, and conditions of depositing the Pt layer that gives conductivity. Even if the same atoms are deposited, in the case of e-beam evaporation, the scattered Pt atoms are regular and the bonding between atoms is strong, so the size of the generated cracks during tension is very large(over 100 um) and the crack size is random. On the other hand, in sputtering, atoms are deposited irregularly and bonding between atoms are relatively weak, so the size of the generated cracks is uniform and conductivity is not easily lost. In this case, in order to control the size of cracks at the nanoscale, the grain size of Pt formed on the surface can be changed by controlling the sputtering time. This plays an important role in determining the size of cracks in tension. As the sputtering time is increased, the thickness and grain size of pt increases. In general, the tensile range of the sensor decreases as the thickness of Pt increases, and the sensitivity of the sensor increases. Depending on the purpose of use, the tensile range and the sensitivity of the sensor can be easily adjusted by controlling the grain size. In addition, when the cracks formed are repeatedly stretched and contracted, the previously cracked part continues to open and close, enabling use without loss of sensor performance. As shown in Figure 2, the stretchable strain sensor based on the nano-crack structure has non-stretchable Pt deposited on the stretchable polyurethane surface. Therefore, when the PU–Pt substrate is stretched by an external tensile force, the non-stretchable Pt on the surface forms a nano-crack and, consequently, the resistance changes. Thereafter, the degree of tensile force can be detected by measuring the resistance. In addition, the flexibility and sensitivity of the sensor can be controlled by controlling the thickness of the deposited metal and the nano-crack structure of the surface. Indeed, the sensor can be manufactured according to predetermined specifications; for instance, it can be designed to collect stable signals with a strain range of 50–150% and a gauge-factor range of 10–30. In other words, both the sensitivity and tension range of the sensor can be controlled to match the strain that occurs at the tip portion of the surgical forceps. This complements the characteristics of vulnerable nano-crack sensors with respect to a wide range of tensile forces. In addition, the sensor can be designed to be applied to human skin, as well as wearable devices that are fixed to either flat or cylindrical surfaces and static or movable surfaces. The fabrication of the flexible substrate is a simple process (Figure 2a). Firstly, thermoplastic-PU beads (Pellethane 2363-80AE; Lubrizol) are dissolved in a mixture of tetrahydrofuran and dimethylformamide with a ratio of 6:4 (*v*/*v*) for one day. Then, the substrate is spin-coated on a slide glass at 100 rpm for 100 s and dried in a 60 °C oven for roughly one day. The substrate can be stretched by up to 300% of its original size. Next, in order to collect electrical signals from both fine and wide tension forces, platinum is deposited on the polyurethane substrate for 100 s by a magnetron-sputtering method with a current of 20 mA at 5 MPa of Ar gas.

### 2.3. Design of Smart Forceps

Creating a system to measure the magnitude and direction of tension is simply not possible with sensitive sensors alone. Rather, the correct material should be used where the sensor is attached; moreover, sensor customization is required. In this case, since the part in question required a complicated shape and sufficient deformation ability, 3D printing was preferred over general mechanical machining. Among the existing 3D-printer materials, VeroClear, which has a fine resolution of about 100 um and positive deformation abilities, can be used to fabricate complex shapes (E: 14.05 GPa). Before fabrication, each part was designed and assembled using SolidWorks. The degree of deformation under constant tension was simulated using COMSOL Multiphysics. Solid mechanics was the condition for COMSOL Multiphysics, while previously created geometry was utilized for SolidWorks. The part that was not deformed was fixed and the deformation was calculated after applying a force of 0.3 N in the *x*-axis direction of the tip. In this case, the deformable part used a custom material with the same Young’s modulus and Poisson’s ratio as VeroClear; the minimum unit of the analyzed mesh was 0.1 mm.

### 2.4. Circuit Configuration for Simulation Data Acquisition

The real-time resistance data of the tension-measurement system was measured using a 12-bit analog-to-digital converter and a PCB circuit composed of a Micro controller unit (MCU) and a light-emitting diode (LED). The resistance-measurement circuit calculates the real-time variation of the strain-sensor resistance, which consists of reference resistance and strain-sensor resistance, using the voltage-divider equation in four independent series circuits. To measure the precise real-time variation of four resistance values, we measured the voltage across the reference voltage using the MCP3204 12-bit A/D Converter Quad Channel SPI Serial IC from Microchip Technology Inc., Chandler, AZ, USA, with a 12-bit resolution. The measured values were calculated as resistance values through ATmega8L, an 8-MHz MCU from Atmel. The entire PCB circuit was manufactured to a size of 25 mm × 40 mm using SMD parts capable of hand soldering.

### 2.5. Evaluation Setup for Nano-Crack-Based Sensor

A single stage and a load cell were set up for accurate movement and measurement with respect to sensor evaluation. To evaluate sensor performance, both ends of the fabricated sensor were attached to a micro-translation stage (M-112; Physik Instrumente, Karlsruhe, Germany) and a load cell (Model UU; Dacell, Korea). At this point, the sensor was deformed by controlling the movement of the micro-translation stage. The applied tensile force was easily obtained via the measured data of the load cell. In order to detect changes in conductivity and resistance, the current passing through the PU–Pt membrane was measured while applying a constant voltage using a source/measure unit (B2902A; Keysight, Santa Rosa, CA, USA). Finally, we used LabVIEW to control the experimental steps, obtaining the current and load data from the source/measure unit and the load cell in the process. In doing so, it was possible to monitor sensor deformation in real time by collecting data for load and conductance changes. An analog-input module for data acquisition (NI 9205; National Instruments, Austin, TX, USA) was used to accurately measure the data of each sensor.

## 3. Results and Discussion

### 3.1. Pre-Strained Method and Nano-Crack-Based Sensor

It was necessary to evaluate the performance of the sensor used in the system, which we did by measuring the fine-strain rate. In order to do this, the sensor was attached to the shaft of the forceps with a pre-tension of 30%. There were three main reasons for this [3]. Firstly, when a sensor is attached to a circular structure, it can become detached. This problem can be overcome by applying pre-tension, which ensures the sensor is fixed in close contact with the circular structure and reduces any noise due to external movement. Second, when the tension is applied, the deformation of the shaft is almost no less than 3%, so the attached strain sensor should be as thin as possible and be able to sufficiently measure the low deformation. For this reason, the thickness of the sensor should be minimized by stretching, according to the Poisson law. This, in turn, maximizes the accuracy of deformation measurements. Indeed, this improves the accuracy of tension measurements on the sensor tip. Thirdly, when pre-tensioned with a 30% strain, data can be obtained both when the sensor stretches and when it retracts. In the case of a circular structure, a total of three sensors should be attached. If the structure then deforms to one side, the resistance of the sensor corresponding to the direction increases, while that of the sensor attached to the opposite side decreases. If the sensor is attached without any pre-tension being applied, such data cannot be collected. Therefore, by attaching a pre-tensioned sensor, accurate and appropriate data relating to direction and magnitude can be obtained.

In order to investigate whether the fabricated sensor can exhibit a resistance change according to a specific strain, a performance evaluation was conducted in the pre-strained working range. Figure 3a shows the resistance changes when the target elongation was increased by 3% increments at 30% strain. As the strain is applied through the micro-translation stage, the relative resistance changes (almost) coincide. The gauge factor corresponding to the sensitivity is 6.924. Next, Figure 3b shows the linearity of the resistance changes and response times of the sensor when the sensor was immediately stopped after applying 0–30% strain. In doing so, it was confirmed that the sensor has a linear resistance change of up to 30% strain, a fast response time of less than 30 ms, and high linearity up to 30% strain (R^2^ = 0.9996). In addition, it performs positively over a wider range than existing sensors with a strain change of less than 3%. This means that a sufficiently stable signal can be obtained even when the sensor is attached to a surgical device. Figure 3c,d show a wired nano-crack sensor and a scanning electron microscope (SEM) image. The size of the generated cracks is uniform without particularly large cracks formed. Currently, cracks are closed due to shrinkage, but it can be expected that the cracks will open as they are stretched and the resistance will gradually increase. Here the sensor is thin and flexible with a thickness of 100 um. Accordingly, it can be attached to the majority of the surfaces. In the SEM image, on the sensor surface, we can see wave-like cracks that are several tens of nanometers in size.

### 3.2. Improvement and Optimization

Commonly used tweezers are larger than general sutures and thus cannot hold thin sutures. Therefore, we designed forceps with a structure similar to that of existing vitreoretinal forceps. The schematic design is shown in Figure 4a. As shown in Figure 4a, the overall system consists of a body made of aluminum, a polymer part that deforms when tension is applied, and a miniaturized PCB part that collects data in real time. The aluminum part, which occupies most of the body, operates on the same principle as conventional vitreoretinal forceps. In detail, when the handle of the middle of the body is pressed, the handle pushes the bar inside the body upward. It is designed to push the polymer-based shaft up and eventually close the gripper. When the gripper is closed, the force is concentrated at the tip, which can hold thin sutures or membranes. Bearings are added between each part to achieve smooth motion. It is necessary to concentrate the contact force in order to hold thinner sutures and membranes well. In order to satisfy this condition, the tip has a complicated shape and a size smaller than 2 mm. Such a shape and size can be easily manufactured using VeroClear. Veroclear is a flexible UV-cured material that can realize a fine shape and can expect a deformation by fine force [26]. With this 3D-printer material, we can manufacture a thin cylinder structure (outer diameter of 4 mm and inner diameter of 3.5 mm) that can be deformed by tension. Since VeroClear is a 3D-printer material, the surface of the final product will not be flat, which is to say it will not provide sufficient frictional force to hold the suture by narrow contact area. In Figure 4a, we can see the principle that allows the tip of the gripper to pick up an object. When the below shaft is moved up and down, the gripper closes. The resulting force is minimal since the gripper does not apply pressure like normal forceps. Rather, it relies on surface friction. Accordingly, in order to solve this friction problem, the friction force is increased by coating the tip with PDMS. Next, when tension is applied to the device, the sensor can measure changes so long as sufficient deformation is present along its direction. In order to attach the nano-crack-based sensor while maximizing the strain, the shaft was formed into a hollow cylindrical shape with an outer diameter of 400 um and an inner diameter of 350 um. Thereafter, we used COMSOL Multiphysics to identify the region with the largest strain when a tension was applied to the gripper. In this case, a tension of 0.3 N was applied to the end of the gripper and the hollow cylinder. In doing so, therefore, we were able to identify the section with the largest deformation.

### 3.3. Position Optimization of Nano-Crack-Based Sensor for Highly Sensitive Detection

The degree of deformation of the sensor attached to the system is proportional to the tension applied to the suture. In corneal surgery, however, the tension is fine (below 0.3 N), so the deformation of the sensor is also fine (roughly 2.3%). Therefore, it is important to improve the performance of the designed sensor, but it is also necessary to optimize the design of the deformation position, the number of attached sensors, and the wiring.

Among these, when deformation occurs, it is possible to confirm the most deformed part through FEM simulation shown in Figure 5 (COMSOL Multiphysics). The unit of figure is von Mises stress (N/m${}^{2}$) that represents the torsional energy at each point. After importing the upper part of the device, a force of 0.3 N, which is the suture-breaking force, is applied in the *x*-axis direction. As a result, the shaft tilts considerably in the x-direction, with an inclination of 7.053, which causes about 2% strain change on the outside of the shaft. In order to maximize the signal-to-noise ratio for the strain, one should identify the region where deformation is maximized in the simulation and where deformation is maximized at the bottom of the shaft. When tension is applied to the tip of the forceps, the shaft deforms in the same direction. However, when deformed, the shaft comes into contact with the underlying structure and, at this point, the level of deformation appears to be greatest. Therefore, it is best to attach the sensor to this part. However, when gripping or releasing the suture, the shaft moves up and down here. Accordingly, it is more appropriate to attach the sensor 5 mm above this part rather than below it as not to disturb the movement. Pre-tensioned nano-crack-based tension-measuring sensors were attached to the shafts, so that both shrinkage and tensile force could be precisely measured as the strain changed. Accordingly, several sensors were attached to precisely measure the tension direction. Theoretically, only one sensor is required for each axis; however, a total of three sensors were attached to ensure sophisticated measurements were obtained (Figure 6a). As a result, resistance data could be intercompared when the sensor was successfully attached, and the size and direction of the tension could be accurately predicted; moreover, stable signal acquisition was possible. Since the shafts are small (roughly 4 mm in diameter), we used a pre-wired sensor. For this purpose, twisted wires (with an inside diameter of 0.25 mm) and silver epoxy were used.

### 3.4. 3-Axis Strain Measurement and System Integration

In this section, the system is verified by measuring the signal along each axis and performing the simultaneous three-axis-measurement experiment in order to evaluate the forceps developed herein.

For this purpose, the force and resistance changes of each axis were measured when the forceps were pulled in each axis direction. In this case, before attaching the three sensors, only one was attached to test whether any resistance changes were present according to the tension. Figure 7a,b show the load and resistance changes when the tensile force was applied in the *x*- and *y*-axis directions. The direction of the applied tension is called the +direction. The suture was repetitively tensioned within 0.3 N of the target tension in the corresponding direction, with sufficient resistance changes being observed. Therefore, we were able to prove that the tension of very thin sutures can be accurately measured, that it is possible to obtain an accurate signal by using three sensors, and that these techniques can be used in corneal surgery. However, in the case of the *z*-axis, the range of the tensile force showing resistance change is as large as 3 N. The deformation of the cylinder is not large enough to exhibit sufficient strain when the shaft is stretched in the same direction. Therefore, a larger tension is required. In terms of the polymer used, several advantages are evident, such as sufficient deformation with the application of fine force. However, breakage can easily occur even with a small force. Therefore, it is necessary to test the durability limit of the polymer in the upper part of the forceps. For this purpose, force was applied until breakage occurred, the results of which are shown in Figure 7d. Here, we can see that the breaking force is roughly 3.5 N. Moreover, the sensor was able to detect sufficient resistance changes until breakage occurred. Based on this, a system was developed that can measure the resistance of each axis in real time. It is necessary not only to collect resistance data through the nano-crack-based sensor but also to transform strain, such as strain or tension along each axis. The PCB, which is attached separately at the bottom of the device, measures the resistance data in real time; it is designed to warn the human (or the robot) by sound or LED signal when the deformations exceed the acceptable standard. The PCB receives resistance from the three sensors and calculates deformation, from which deformation direction and tension force can be calculated.

Initially, we tried to process the data with an algorithm that calculated the resistance data of the sensors corresponding to the three axes as deformations for each independent axis. However, when tensioning with one axis, the sensor corresponding to the other axis also has a change in resistance, so there is a limit just to collecting data individually. Since it is not easy to determine the magnitude and direction of the tension in this way, we have added an algorithm that automatically compensates for the deformation and direction of the other axes according to the change in tension to one axis. Figure 8 shows the results of applying tension to the left, right, and bottom, respectively, using an integrated tension-measuring system after attaching a suture to a load cell. Generally, the load data measured in the load cell represents the numerical magnitude of the force, not the direction. However, in the case of the developed system, the resistance changes according to the tension are measured and converted into strain rate and direction with respect to the x- and y-coordinates. Accordingly, the tension direction and magnitude can be visualized in real time. Unfortunately, the exact error rate cannot be measured because the sutures are pulled by hand; however, it was confirmed that deformation was sufficiently measured and visualization was possible in real-time. This means that when the robot moves the forceps at a certain level and direction, it is possible to check whether the movement has been accurately performed and the applied tension can be checked for resistance change. This enables more accurate movement and feedback when there is a problem with movement.

### 3.5. Ex-Vivo Test in Integrated System

A fine tension-measuring forceps system has already been designed as a step in the development of basic technology for automation and robotic surgery, both of which require sophisticated suturing. However, in surgery, complex movements are needed, such as twisting or knotting the threads rather than simply pulling them. In order to achieve this, the corneal-suture experiment was performed with pigs’ eyes using the developed system in Figure 9. Eyes within 3 days after slaughter were used and the cornea was separated in advance. Using the developed forceps, a suture to which the needle was connected was held and the cornea separated in advance was sutured to the pig’s eye. In order to hold the suture and perform complex movements, it is necessary to hold and control the suture reliably. Since it is larger than a conventional surgical tool, it is important to check whether forceps can hold the suture and have precise control. As a result of the experiment, it was confirmed that although it is not possible to move completely finely, it is possible to control enough to be able to suture. However, the sophistication is less than that of the existing forceps and this must be improved in the future. When performing surgery on the eyes, tension was measured by pulling the suture after tying the knot. In addition, the system-verification experiment was conducted using the LED signal (Figure 10). As a result, it was possible to hold the suture when tension was applied after tying the knot. When the tension was increased to exceed the standard tension, the LED signal light turned red.

## 4. Conclusions

Automated robotic surgery and conventional sensnrs are not applicable to corneal surgery, which requires fine-force control. In this paper, a new surgical instrument integrated with a flexible nano-crack sensor was proposed. This instrument is an integrated system that can measure the microscopic deformations that occur when sutures are pulled and flexible nano-crack sensor is designed to be miniaturized and applicable to various structures. The nano-sensor can be designed according to the required specifications for sensitivity and tensile range. Throughout the experiment, the nano-crack sensor demonstrated high sensitivity (GF = 6.924) and great sequential linearity (R${}^{2}$ = 0.9996) with required working range (30% strain). As a result, the resistance change could be observed at a tensile force of 0.3 N or less. After then, Three nano-crack-based sensors are attached to the forceps and the resistance data from each sensor was automatically collected by the signal-processing system. The relationship between the transformed resistance and the material being deformed could be used to calculate the strain and tension along each axis. To test the developed device, a successful suture test was performed using pig eyes. However, VeroClear used in the experiment is not biologically dangerous and resistant to heat, but it will require verification related to sterilization and disinfection in order to be used in surgery. However, in the process of fusing data from multiple sensors, there is a lack of a system regarding the direction of the axis on which the force is applied, and while the biaxial strain calculation for shear force can be accurately measured, the *z*-axis strain needs further improvement. However, since most corneal surgery only uses shear force to suture, it is not a problem in practical application. Also, when no tension is applied, resistance change due to deformation of the flexible layer is observed, which affects to the accuracy of the sensor. This can be improved with a feedback method that measures tension as a rate of change in resistance. In addition, proper embedding and packaging of sensor systems is very important with regards to hygiene and commercialization. This study has developed a tension-measurement system that can be used in current corneal surgery or in future automated surgical robots. In the future, overall performance can be improved by allowing for precise sensing and accurate tension-vector measurements. Based on this study, resistance collection and feedback systems from forceps tips will minimize tissue damage due to human error, such as hand slippage or shaking.

## Figures and Tables

**Figure 1 polymers-13-04433-f001:**
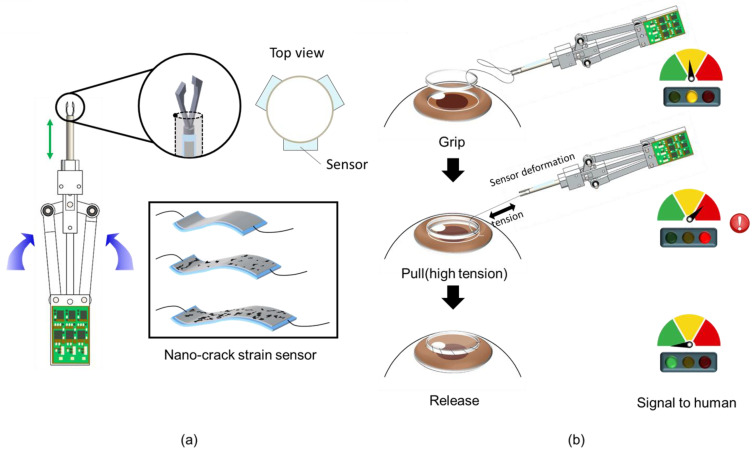
Overall schematic of tension-measurement system for forceps. (**a**) Overall configuration, which is divided into the tip part, the system body, and the PCB. Press the handle on the side to raise the shaft and grab the suture. Nano-crack sensors are attached to the shaft surface. (**b**) Schematic of corneal surgery using forceps. We can hold the suture with forceps and proceed with corneal surgery. At this time, the tension of the suture is measured and this can be visually confirmed.

**Figure 2 polymers-13-04433-f002:**
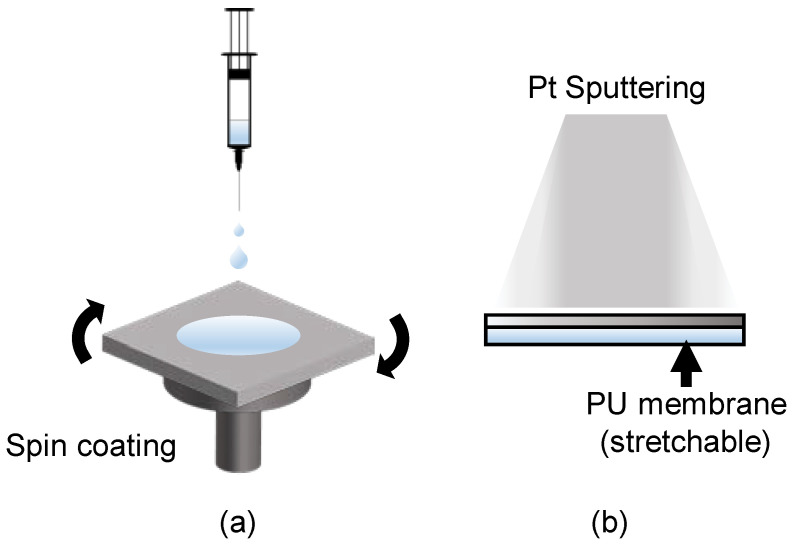
Fabrication process of the PU–Pt membrane. (**a**) Base layer fabricated by spin coating polyurethane. (**b**) Deposition of the platinum layer on spin-coated polyurethane membrane by Pt sputtering.

**Figure 3 polymers-13-04433-f003:**
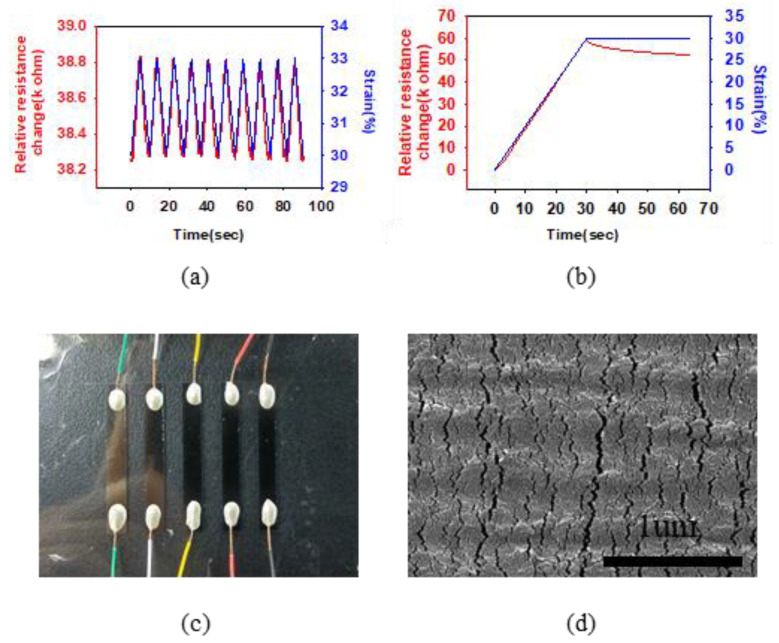
Performance of nano-crack sensor. (**a**) Relative resistance change of the nano-crack sensor according to 0–50% strain. (**b**) Response-time test of nano-crack sensor. (**c**) Nano-crack sensor. (**d**) SEM image of nano-crack sensor.

**Figure 4 polymers-13-04433-f004:**
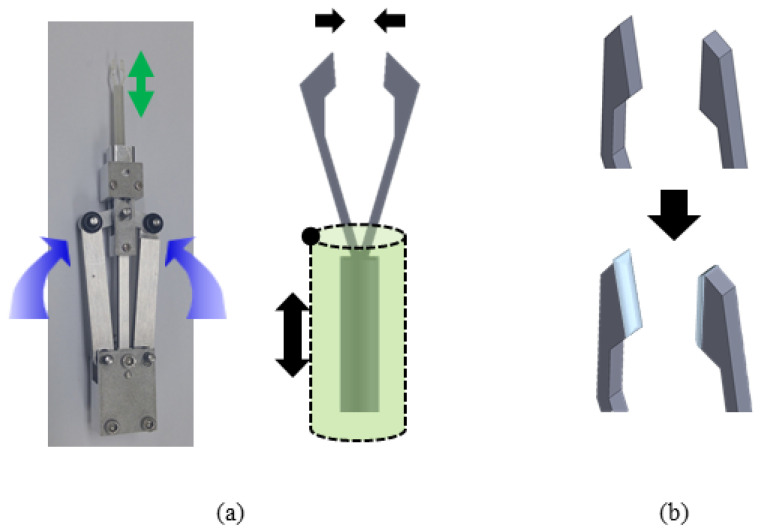
Working principle of the tension measuring system. (**a**) Pressing the handle on the side raises the shaft with the bar inside the system. After that, the shaft can close the gripper and hold the suture. (**b**) Coating polymer to the tip to increase friction.

**Figure 5 polymers-13-04433-f005:**
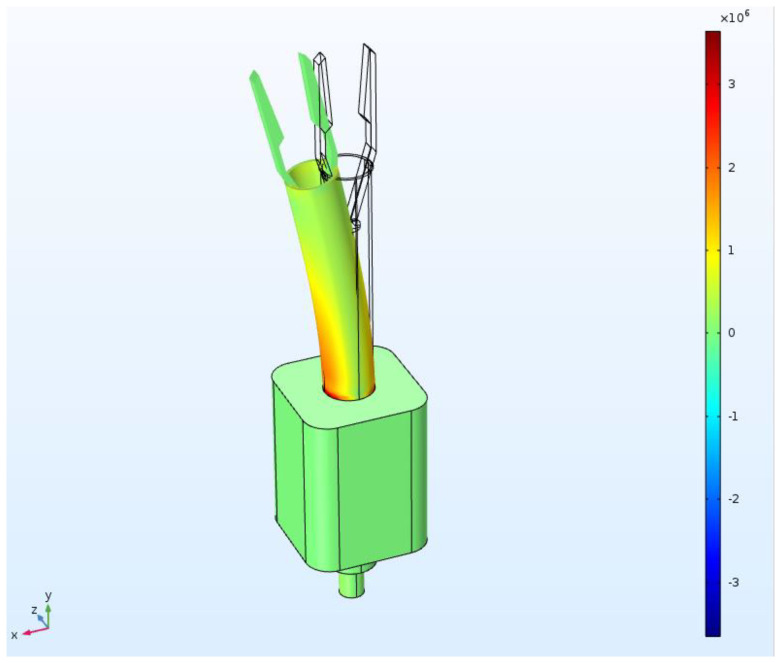
Deformation-rate simulation when tension is applied to the tip of the forceps.

**Figure 6 polymers-13-04433-f006:**
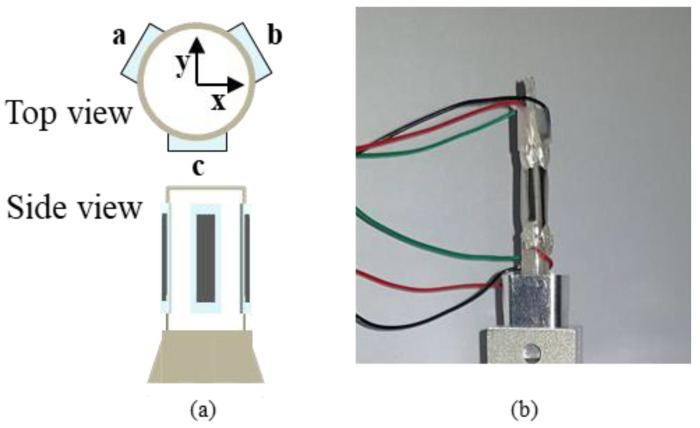
Sensor attachment position design. (**a**) Top and side view of the sensor attachment to the shaft. (**b**) Sensor fully attached to the shaft and wiring.

**Figure 7 polymers-13-04433-f007:**
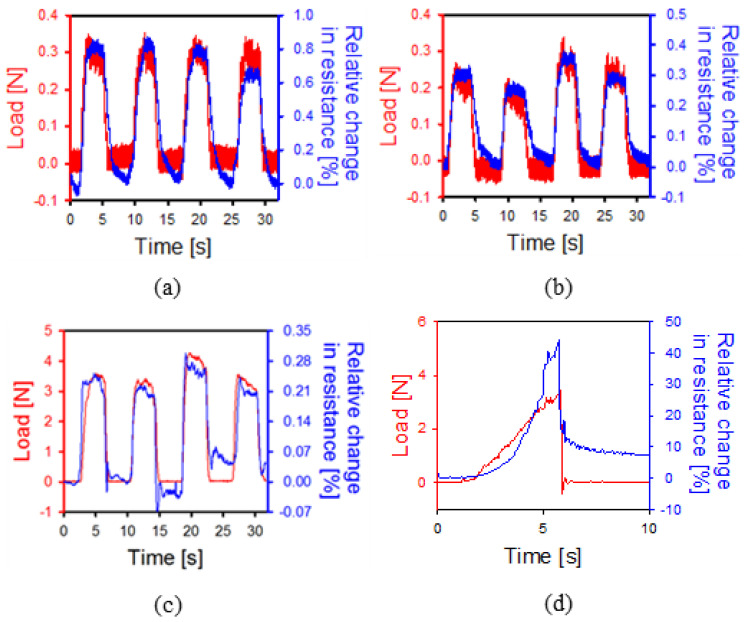
Strain measurements for each axis using the tension measurement system. (**a**) X-direction. (**b**) Y-direction. (**c**) Z-direction. Deformation is not well detected in the same axial direction as the shaft. (**d**) A load that can endure suture failure.

**Figure 8 polymers-13-04433-f008:**
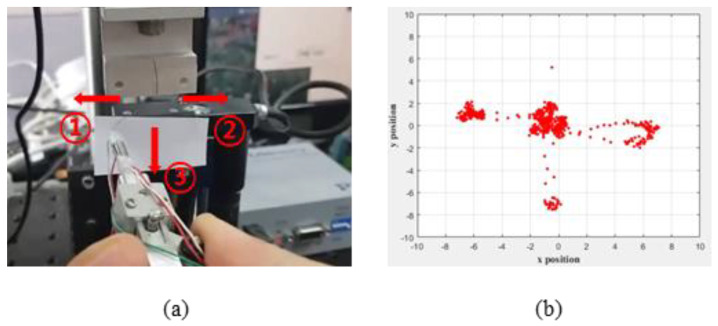
Real-time integration direction and tension-magnitude measurement using tension-measurement system. The device has three sensors. An algorithm has been added that automatically measures the three resistances and converts them into a single direction and magnitude. (**a**) Experimental process. (**b**) Tracing the signal according to direction and deformation rate caused by tension.

**Figure 9 polymers-13-04433-f009:**
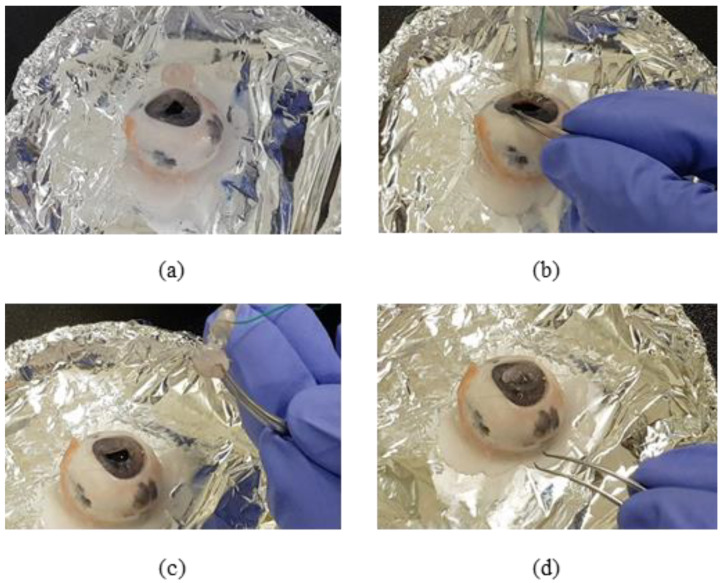
Pig-eye suture surgery. Surgery was performed using the developed sensor system. (**a**) Corneal extraction. (**b**) Pig-eye stitching. (**c**) Pig-corneal stitching. (**d**) Eye and corneal suturing.

**Figure 10 polymers-13-04433-f010:**
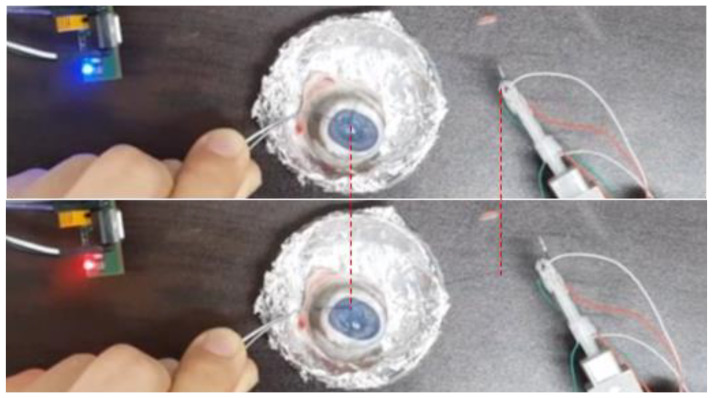
Confirmation of warning function according to tension with operation of pig-eye knot. Before (blue) and after tensioning (red).

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
