# Peer review of "Three-Axis Tension-Measuring Vitreoretinal Forceps Using Strain Sensor for Corneal Surgery"

_polymers, 2021, doi:10.3390/polym13244433_

Round 1

Reviewer 1 Report

This manuscript presents the idea developing process of a nano-crack sensor based tension-measurement system for the monitoring of suture holding force on the forceps used in corneal surgery. It has potential reference value in the development of precise motion control in robotic surgery. However, the whole context seems to be rough at present writing, not only some important technological points having no revealed quantitatively but also the writing normative.

E.g., How could adjust the surface nano-cracks to the desired size, and what relationship between the crack sizes and the sensor sensitivity and measurable tensile range? And how could keep the nano-cracks constant and the deposited Pt particles no detachment under repeated stretching?

Note that the reference number starts from [5] as the first one and the following not in a natural order in the text may be ill-formed. Also note the order number: 1. Results and discussion and 1.1. Pre-strained method and nano-crack-based sensor? In the caption of Fig.1, there is lack of the information for (b). The schematic drawing of Fig.2 (a) presents just the spin-coating of polyurethane membrane, not the deposition of the platinum layer on it. The schematic drawing of Fig.2 (b) does not match the caption meaning also. Line 87, “In order words” may print error. Line 180, “This is an example of a quote.” What’s the example?

More important, its theme discussed therein, though more or less involving the use of polymers including suture, largely oriented on sensor device, thus it may be more suitable to publish other sensor related journals.

Author Response

Thanks for the thoughtful comment.

All comments about the format have been corrected, but among them, "in other words" in Line 87 was not changed because it was thought to be an appropriate term.

In the case of nano-cracking questions, we have added to the text about the crack generation and grain size, and the resulting sensitivity and tensile range. (Line 118-130)

To quote what was added:

In order to make a nano-crack sensor, it is necessary to control the crack size of the sensor surface. The size of the crack depends on the method, time, and conditions of depositing the Pt layer that gives conductivity. Even if the same atoms are deposited, in the case of e-beam evaporation, the scattered Pt atoms are regular and the bonding between atoms is strong, so the size of the generated cracks during tension is very large(over 100um) and the crack size is random. On the other hand, in sputtering, atoms are deposited irregularly and bonding between atoms are relatively weak, so the size of the generated cracks is uniform and conductivity is not easily lost. In this case, in order to control the size of cracks at the nanoscale, the grain size of Pt formed on the surface can be chaged by controlling the sputtering time. This plays an important role in determining the size of cracks in tension. Depending on the purpose of use, the tensile range and the sensitivity of the sensor can be easily adjusted by controlling the grain size.

The edited content is attached as an attachment.

Reviewer 2 Report

Authors selected interesting research problem, it really needed by the medical world with proper guidance and scientific awareness. There are following comments for further improvment of the paper.

Major Comments

  1. What are strengths of nano-crack sensor , and how these are different from Strain sensors? It shall be great if authors give the differences between sensor nodes with their limitations and advantages in robotics surgery. Why nano-crack sensors are appropriate and highly required for corneal surgery ?
  2. What is interconnection between Vitreoretinal Forceps, sensor sensitivity, and strain rate in robotic surgery ? Authors are highly recommended to sheds some more light on these relationships.
  3. Research gaps, objectives of the proposed work should be clearly justified.
  4. To improve the Related Work and Introduction sections authors are highly recommended to consider these high quality research works <‘A Novel Adaptive Battery-Aware Algorithm for Data Transmission in IoT-Based Healthcare Applications, Electronics, MDPI, Vol.10, No.4, pp.367, 2021'>, <A Multi-sensor Data Fusion Enabled Ensemble Approach for Medical Data from Body Sensor Networks’,Information Fusion, Elsevier, Vol.53, No.2020, pp.155-164, 2020>
  5. Authors must explain in detail the introduction section.
  6. English must be revised throughout the manuscript.
  7. Limitations and Highlights of the proposed methods must be addressed properly
  8. Experimental results are not convincing, so authors must give more results to justify their proposal.

Major changes are recommended 

Author Response

Thanks for the thoughtful comment.

The reason why the current commercial sensor cannot be used and the reason why the nano-crack sensor has strength in this case are added. Added information on how it affects the strain rate and tensile range. (Line 118-130) Using this, a sensor with a target range and sensitivity can be designed, and it can be sufficiently applied to corneal surgery.
However, for the proposed technique, sufficient prior studies do not exist and comparative studies are lacking. Limitations of the proposed method were added in the conclusion (Line 388). In the case of the presented results, the direction and size of the tensile force applied to the suture were successfully measured, and although it is not perfect, because the suture is very thin at 100 um, but we think it will be help in future related research.

The revised content will be sent as an attachment.

Reviewer 3 Report

In this paper, the authors developed a new tension-measurement system for forceps used in corneal surgery that could realize precise motion control and visualize the collected date. This paper has a certain degree of novelty and has illustrated the design and methods of the system in detail, however there still exist some questions in this manuscript. The following are questions in this manuscript:

  • The text is not well arranged in terms of layout and there is a lack of alignment among parts.
  • In page 6 line 17, the authors mentioned that ‘In the SEM image, on the sensor surface, we can see wave-like cracks that are several tens of nanometers in size’. It is suggested that the authors should illustrate the special meanings or implications of these wave-like cracks.
  • In page 8 line 13, the authors mentioned that ‘Theoretically, only one sensor is required for each axis; however, a total of three sensors were attached to ensure sophisticated measurements were obtained’. Specific number of required sensors ought to be judged by experiments and particular circumstance.
  • In general, the authors had greatly illustrated the theories of the new system, however, it is suggested that authors ought to expand on experiments more in detail.

To sum up, I suggest that this manuscript needs minor revision.

Author Response

Thanks for the thoughtful comment.

The format error has been corrected, and I'm sorry for not paying attention in advance.

Added information on the effect of cracks on sensor sensitivity, tensile range, and grain size (Line 118-130), and added information on the meaning and role of cracks in Fig.3(d)(Line 230-233).

As for the number of sensors used, usually only 2 sensors are needed as the force is applied in the vertical direction of the forceps, but I used 3 to increase the accuracy of the sensors. We are sorry that the number of sensors was arbitrarily set because there was no prior research on tension measurement in related surgery.

The revised content will be sent as an attachment.

Reviewer 4 Report

The manuscript is devoted to the study of a force control transducer for a surgical instrument. This problem is important for eye surgery. a new surgical instrument integrated with a flexible nano-crack sensor was proposed. Throughout the experiment, the nano-crack sensor demonstrated high sensitivity (GF = 6.924) and great sequential linearity (R2 = 0.9996) with required working range (30% strain). In general, a rather narrowly specialized subject area of research is presented.

These research results are original and have scientific value. Nevertheless, there are certain moments demanding explanations.

  1. The purpose, tasks and novelty of the work are not quite clear from the text of the manuscript. What is the main challenge - material selection, sensor design optimization, or sensor calibration? The purpose of the research should be more clearly stated.
  2. It is necessary to indicate the requirements for the sensor: measurement range, sensitivity, error in percent, high-speed performance.
  3. The relationship between the thread tension and the deformation and tension of the forceps rod is not shown. How is this connection established - theoretically or experimentally?
  4. In the course of the presentation of the material, various terms are used: stress, deformation, force. It is necessary to more clearly distinguish between these concepts and bring the relationship between them.

Author Response

Thank you for your suggestion. We agree with the suggestion that the purpose of the study is not clear, and the related information was added in line 37. “The purpose of this study is to develop a forceps capable of precise movement and a system that can measure the tension in the range where the suture does not break in real time.”

In the case of the second and third comments, there isn't clear standard because there is no precedent for applying a sensor system to the form of forceps for corneal surgery. Therefore, experimentally, we arbitrarily fabricated a system that can hold a thin suture with a diameter of 100 μm and measure the tension when the suture pulled. The nano-crack sensor used in this case, the measurement sensitivity and tensile range can be adjusted according to the thickness of the crack. and fig. 5, when the tension generated when the suture is pulled, the shaft is deformed in the corresponding direction, and the sensor attached to the outside of the shaft measures this deformation.

In order to show the tension and stress more clearly in the figure, the part about sensor deformation is shown in figure 1. We also added on line 93 how tension and deformation relationships are formed as an explanation.
"When tension is applied to the suture, stress is applied to the shaft of the forceps by the tension, and accordingly, the shaft is bent and deformed. The tension applied to the suture can be calculated on the principle that a sensor attached to the shaft surface measures the degree of this deformation"

Also, in line 257, although the stress and force are proportional, it was replaced because the force was considered appropriate rather than the stress.” Veroclear is a flexible UV-cured material that can realize a fine shape and can expect a deformation by fine force”

Round 2

Reviewer 1 Report

Line 88 (Line 87 in version 1 ), the short term ‘In order words’ still be in printing error, please the authors have a check.   ---------’tension. In order words, from a kinematic point of view, the forceps should be able to’.

Line 127, the word ‘chaged’?

Though the authors added the Line 118-130 into the text for explaining the questions asked in the last review, but the answer is neither complete nor quantitative, even not targeting the main theme of the questions— ‘How could adjust the surface nano-cracks to the desired size, and what relationship between the crack sizes and the sensor sensitivity and measurable tensile range? And how could keep the nano-cracks constant and the deposited Pt particles no detachment under repeated stretching?’

Thus, I think this version of manuscript is still not qualified.

Author Response

We agree with your assessment. Thank you for your suggestion. Incorrect terminology has been corrected.

Among the previous comments, the details on nanocrack size control can be found in line 130. And we agree that the lack of explanation for the correlation between crack size and sensor sensitivity and tensile range. So we added an explanation for this to line 132. "As the sputtering time is increased, the thickness and grain size of pt increases. In general, the tensile range of the sensor decreases as the thickness of Pt increases, and the sensitivity of the sensor increases."

The reason why the size and number of cracks are kept constant when repeatedly tensioned and contracted is that the cracks created in the first place continue to open and close and keep the cracks constant. An explanation about this has been added to line 136."In addition, when the cracks formed are repeatedly stretched and contracted, the previously cracked part continues to open and close, enabling use without loss of sensor performance."

Reviewer 2 Report

Authors have slightly been improved

Author Response

Sorry if the content is not satisfactory. We are working hard to make it as good as possible, and we will send you a correction.
It was added in line 37 because the purpose of the study was not clear.“The purpose of this study is to develop a forceps capable of precise movement and a system that can measure the tension in the range where the suture does not break in real time.”

There are the lack of explanation for the correlation between crack size and sensor sensitivity and tensile range. So we added an explanation for this to line 132. "As the sputtering time is increased, the thickness and grain size of pt increases. In general, the tensile range of the sensor decreases as the thickness of Pt increases, and the sensitivity of the sensor increases."

In order to show the tension and stress more clearly in the figure, the part about sensor deformation is shown in figure 1. We also added on line 93 how tension and deformation relationships are formed as an explanation.
"When tension is applied to the suture, stress is applied to the shaft of the forceps by the tension, and accordingly, the shaft is bent and deformed. The tension applied to the suture can be calculated on the principle that a sensor attached to the shaft surface measures the degree of this deformation"
Thank you.

Round 3

Reviewer 1 Report

This time the authors give their short explanations for my last questions in the corresponding places of their revised version. Their given reasons might be somewhat reasonable but obviously are in less quantitative data. Thus I suggest it would be better supported if adding some related references. 

Reviewer 2 Report

Paper is improved at some extent, but still there are core issues to be fixed

Comments

  1. Paper lacks detailed methdology section with clear flowchart, datasets used, tools and platforms considered
  2. Also, pseudocde of the proposed method is missing 
  3. Limitations of the proposed work must be highlighted 
  4.